# Optimizing Soil Management for Sustainable Viticulture: Insights from a Rendzic Leptosol Vineyard in the Nitra Wine Region, Slovakia

Vladimír Šimanský [1],*, Elżbieta Wójcik-Gront [2], Jerzy Jonczak [3] and Ján Horák [4]

1   Institute of Agronomic Sciences, Faculty of Agrobiology and Food Resources, Slovak University of Agriculture, 949 76 Nitra, Slovakia
2   Department of Biometry, Institute of Agriculture, Warsaw University of Life Sciences—SGGW, 02-787 Warsaw, Poland; elzbieta_wojcik-gront@sggw.edu.pl
3   Department of Soil Science, Warsaw University of Life Sciences—SGGW, 02-787 Warsaw, Poland; jerzy_jonczak@sggw.edu.pl
4   Institute of Landscape Engineering, Faculty of Horticulture and Landscape Engineering, Slovak University of Agriculture, 949 76 Nitra, Slovakia; jan.horak@uniag.sk
*   Correspondence: vladimir.simansky@uniag.sk

**Abstract:** Properly chosen soil management practices can stabilize the nutrient regime in the soil, including the mobility and bioavailability of hazardous elements. This study aimed to identify the optimal soil management practices in a productive vineyard on Rendzic Leptosol in the Nitra wine region (Slovakia). Soil samples were collected each spring from two depths, 0–30 cm, and 30–60 cm, with the following treatments: T—soil tillage, P + FYM—plowed farmyard manure, G—grass strips, G + NPK1—first-level fertilization, and G + NPK2—second-level fertilization, from 2019 to 2023. The results indicated that more pronounced changes in soil properties occurred in the 0–30 cm layer. Higher NPK rates significantly affected soil sorption capacity and decreased soil pH when compared to other treatments. While G + NPK2 showed the highest storage of total N, S, P, K, and available P and K, it exhibited the lowest levels of total and available Ca. The T treatment displayed the lowest storage of C, N, S, P, and available K. In terms of hazardous metals (Cr, Ni, Pb, and Zn) none exceeded the limiting values in any of the soil management practices. However, in the 0–30 cm layer, Cu concentrations exceeded the limits set by Slovak Republic regulations in the T, P + FYM, G, G + NPK1, and G + NPK2 treatments by 62.6, 73.7, 70.2, 82.1, and 102.9 mg kg$^{-1}$, respectively. Additionally, as total C increased, Cr concentration was observed to decrease with correlation (r = −0.46). Positive correlations were found between total C and Zn, as well as CaCO$_3$ and Zn in the 0–30 layer.

**Keywords:** grass cover; soil amendments; nutrient availability





## 1. Introduction

Soil management practices in vineyards are very important to facilitate the high yields and health of cultivated grapes for high quality wine production. Nevertheless, adhering to the principles of sustainability in vineyard management is an essential prerequisite to adhere to European sustainability goals [1]. In recent times, conservative soil management techniques played an important role in vineyard soils, as opposed to conventional tillage, in efforts to preserve biodiversity, maintain soil fertility, and uphold the vegetative-productive balance [2]. The cultivation of grass between vine rows, often referred to as grassy strips or green strips, offers significant advantages for winegrowers. Even in vineyards with deep soils, Celette et al. [3] recommended intercropping instead of permanent grassing. These grass strips have several documented benefits as they facilitate reduced soil erosion [1,4], enhance water infiltration, increase soil water retention [2,5], improve soil structure and holistically enhance an array of physical properties in vineyard soils [6], and contribute positively to soil C sequestration [7]. Given the context of climate change, well-chosen soil

management in vineyards is critical, as it can help mitigate the challenges posed by alternating periods of prolonged drought and excessive rainfall. When considering the optimal nutrient balance for grapevines through chemical fertilization, there are potential risks such as nutrient overloading, increased availability, and heightened leaching. Chemical fertilization can also alter the soil environment, potentially increasing the mobility of harmful elements and their subsequent absorption by plants and entry into the food chain [8]. As previously mentioned, grassing has its benefits but also disadvantages, especially in Mediterranean regions where water is a limiting factor for plant production [9]. Grass cover competes with vine plants by depleting nutrients and water. Grapevines are relatively demanding regarding nutrients and the soil must maintain a stable nutrient supply to ensure quality and sufficient grape production; however, they can extract them even from less accessible forms [10]. Achieving this without heavy reliance on chemical fertilizers is increasingly attractive to winegrowers, driven by economic considerations [11]. For instance, the growing popularity of organic amendments, such as composts, in vineyards has the potential to significantly influence the availability of essential and beneficial plant nutrients [1].

Soil management practices can alter or modify soil chemistry and soil sorption capacity, subsequently impacting the behavior of macronutrients and hazardous elements within the soil. Considering existing knowledge gaps in this area, this study aims to: 1. evaluate the effect of long-term soil management practices in vineyards on changes in soil sorption capacity, macronutrient content, and hazardous elements and 2. identify relationships between soil sorption capacity and nutrient regimes, including hazardous elements, under different soil management practices in vineyards. We hypothesize that long-term and well-planned soil management practices in vineyards will contribute to a balanced soil chemistry, potentially leading to the stabilization of nutrient and hazardous element content (H1). Furthermore, we anticipate that this effect may be significantly influenced by anthropogenic activities, particularly the application of high doses of NPK fertilization (H2). The presence of grassing is expected to have a positive effect on the mitigation of leachable hazardous elements; and the stabilization of nutrient regimes and soil sorption capacity (H3).

## 2. Materials and Methods

### 2.1. Site Description

The study was conducted in a vineyard situated in Nitra-Dražovce (48°21′6.16″ N, 18°3′37.33″ E) in Slovakia, during the years 2019–2023 (Figure 1). Nitra-Dražovce is located to the northwest of Nitra city within the Nitra wine-growing region of Slovakia, and the experimental vineyard is situated in the southwestern foothills of Zobor Hill, primarily composed of Mesozoic sedimentary rocks, including Cretaceous, Jurassic, and Triassic limestone [12]. The climate in Nitra-Dražovce falls under the mild continental category, characterized by warm summers and full humidity, conforming to the Cfb classification in the Köppen climate system [13]. The most recent 30-year normal data (1991–2020) for Nitra-Dražovce indicate an average annual precipitation of 550 mm and a mean temperature of 9.8 °C. Before the start of the experiment, the soil was classified as Rendzic Leptosol by the IUSS Working Group WRB [14]. In the 0–30 cm soil layer, 8% rock fragments, 56.9% sand, 33% silt, and 10.1% clay were the major solid fractions. Its bulk density measured 1.29 g cm$^{-3}$. The soil had 17 g kg$^{-1}$ of soil organic carbon, 1067 mg kg$^{-1}$ of nitrogen, 99% base saturation, 99 mg kg$^{-1}$ of available phosphorous, and 262 mg kg$^{-1}$ of available potassium. On average, the soil pH was found to be neutral at 7.2.

### 2.2. Experimental Setup in the Vineyard

The vineyard on which the experiment with different soil management practices was established and is still ongoing is located on the southwestern slope of Zobor Hill. In the 11th century, the southern slopes of Zobor Hill were deforested, and vineyards were planted. Today, the locality is used as a horticulture area and for growing plants to produce wines [15]. In the autumn of 2000, the site was plowed to a depth of 30 cm, and

vines (*Vitis vinifera* L. cv. Chardonnay) were planted in rows at a spacing of 2 × 1.2 m (th eRheinish—Hessian system was used). During the post-establishment phase (2000–2006), the vineyard was managed intensively by mechanically removing weeds three times a year on average, and all inter-rows were plowed to a depth of 25 cm in the autumn. Full productivity was achieved in 2006, coinciding with the onset of this experiment. The experiment was designed to contrast the effects of five different soil management practices in a productive vineyard and was laid out as a block design with three replicates. Experimental plots are 3 m wide and 7 m long to cover a row of 7 vine plants. The experiment involved following different soil management practices.

1. **Grass strips (G)**: In this practice representing a control treatment because it involves the smallest or no human intervention (mechanical) in the soil environment during the experiment. A mixture of grasses, including *Lolium perenne* L. (50%), *Poa pratensis* L. (20%), *Festuca rubra* subsp. *commutata Gaudin* (25%), and *Trifolium repens* L. (5%), was sown between the vine rows. Plant cover is maintained in and between the vine rows by mechanized mowing 4 times a year on average, and by leaving all cuttings in situ as mulch. No fertilization is applied.

2. **Tillage (T):** Tillage represents intensive vineyard management and a typical wine-grower model in this area. Soil tillage in the interrow of the vine involved annual plowing in the autumn to a depth of 25 cm. Manual weeding between the vine rows using hoes is carried out during the growing season as needed, typically about three times per season. No fertilization is applied.

3. **Plowed farmyard manure (P + FYM):** This practice included autumn plowing to a depth of 25 cm annually. Additionally, farmyard manure was plowed into the soil at a rate of 40 t ha$^{-1}$ in a 4-year cycle, with applications in 2005, 2009, 2013, 2017, and 2021. Poultry manure was used, containing 55% organic substances, 2.8% Nt, 1.3% P pentoxide ($P_2O_5$), and 1.2% K oxide ($K_2O$) in dry matter, with a pH range of 6 to 8. Similar to the T treatment, soil tillage of the interrows was performed mechanically around three times during each growing season.

4. **Grass strips and NPK fertilization at the first level (G + NPK1):** This practice involved the application of 100 kg ha$^{-1}$ N, 30 kg ha$^{-1}$ P, and 120 kg ha$^{-1}$ K. The NPK dose was divided, with half applied in the spring (bud burst—in March) and the other half during flowering (in May). Grass biomass was cut down around three times during the vine's growing season (remaining on the soil surface as mulch).

5. **Grass strips and NPK fertilization at the second level (G + NPK2):** This practice included the application of 125 kg ha$^{-1}$ N, 50 kg ha$^{-1}$ P, and 185 kg ha$^{-1}$ K. Like the G + NPK1 practice, the NPK dose was divided, with two-thirds applied in the spring and one-third during flowering. As with the other practices, grass biomass was cut down around three times during the vine's growing season.

In G + NPK1 and G + NPK2 treatments, annual fertilizer application (Duslofert Extra NPK(S) 14-10-20-7) at 20 cm depth is used. Duslofert Extra is a complex granular fertilizer that contains the basic nutrients nitrogen, phosphorus, potassium, and the secondary nutrients sulfur, calcium, and magnesium. Nitrogen is in nitrate and ammonium form, phosphorus is in water-soluble form, and potassium is in sulfate form. The content of chlorides is less than 2% by weight, the content of total nitrogen (N) 14%, the content of nitrate nitrogen 5.7%, the content of ammonium nitrogen 8.3%, the content of total phosphorus oxide ($P_2O_5$) 10%, the content of phosphorus oxide ($P_2O_5$) soluble in neutral ammonium citrate 9.5%, content of phosphorus oxide ($P_2O_5$), soluble in water 9.0%, content of total potassium oxide ($K_2O$) soluble in water 20% and content of sulfur (S) soluble in water 7%.The fertilization rate follows the manufacturer's recommendation for two commercial yield levels of 10 and 15 t ha$^{-1}$ of grapes a year in Slovakia [16].

All treatments in the vineyard were uniformly treated against diseases and pests throughout the duration of the experiment. Significant damage caused by pests was not observed and therefore insecticides were not used in significant doses. The following preparations were used to protect against fungal diseases: based on copper hydroxide—Cuprozin

Progress, Funguran Progress, and Champion 50 WG. Based on copper hydroxide + copper oxychloride—Airone SC, and Coprantol Duo. Finally, based on the active substance Mancozeb (contains zinc)—Dithane M45.

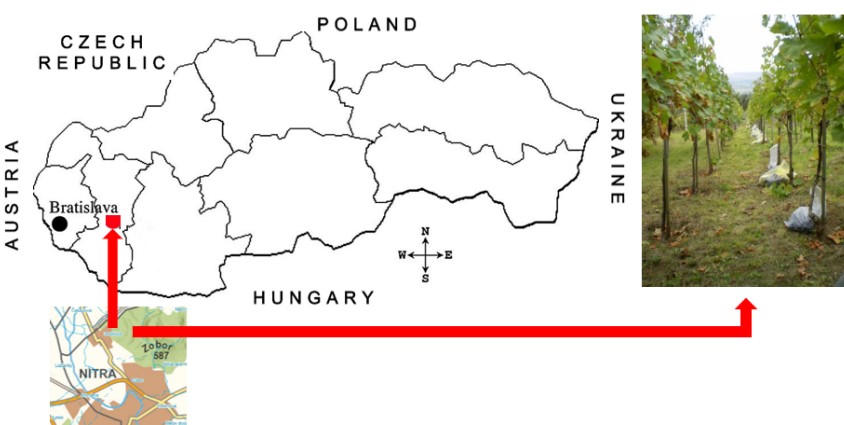

**Figure 1.** Experiment vineyard location.

### 2.3. Soil Sampling and Analysis

During 2019–2023, soil sampling was carried out at the soil depth of 0–30 cm and 30–60 cm in spring. Sampling included all treatments of soil management practices in the productive vineyard. Each treatment (21 m$^2$) with three replicates included three sampling points. Finally, in each replicate, an average soil sample was created by a mix of 3 sampling points. In total, 75 soil samples were taken during the entire period. The collected soil samples were subsequently homogenized, dried, ground, and sieved through a 0.25 mm mesh sieve. The following soil characteristics were determined: soil pH in H$_2$O (a soil-to-distilled-water ratio of 1:2.5), determined potentiometrically using a pH meter (HI 2211, HANA Instruments). Parameters of soil sorption capacity such as hydrolytic acidity and sum of basic cations were determined using the Kappen method [17]. Analytical determined hydrolytic acidity (Ha) by titration in a 1 mol L$^{-1}$ CH$_3$COONa extract and sum of basic cations (SBC)—titrated in leachate with 1 mol L$^{-1}$ HCl were used for calculation of cation exchange capacity (CEC) according to Equation (1).

$$CEC = Ha + SBC, \tag{1}$$

Base saturation (Bs) of the soil colloidal sorption complex was calculated according to Equation (2).

$$Bs = SBC/CEC·100, \tag{2}$$

Carbonates were determined by volumetric method using a Jankov calcimer, based on the CO$_2$ evolution after reacting with HCl (diluted with water in a 1:3 ratio). Total carbon (C), nitrogen (N), and sulfur (S) were analyzed by dry combustion (Vario MacroCube, Elementar, Germany). The content of available forms of phosphorus (P), potassium (K), and calcium (Ca) were determined by inductively coupled plasma atomic emission spectrometry (ICP-OES, Avio 200, Perkin Elmer, Waltham, MA, USA) before soil extraction according to Mehlich III procedure [18]. The contents of close to total forms of chromium (Cr), copper (Cu), nickel (Ni), lead (Pb), and zinc (Zn) were analyzed by ICP-OES (Avio 200, Perkin Elmer, USA) before microwave digestion (Ethos Up, Milestone, Milan, Italy) in aqua regia. Only high purity reagents were used for sample extraction and digestion. Quality of analysis was ensured by using certified reference materials (MERCK).

### 2.4. Statistical Analysis

A three-way analysis of variance (ANOVA) was conducted using Statistica (Statistica Ver. 13) software to assess the overall variability in the dependent variables attributed to

Tillage, Plowed FYM, Grass strips, G + NPK1, G + NPK2, sampling depth, and the sampling year, years from 2019 to 2023. The comparisons of treatments were presented separately for each layer (0–30 cm and 30–60 cm). The null hypothesis posits that there are no significant distinctions between the different treatment conditions, sampling depth or years. The dataset adheres to ANOVA assumptions, which include the independence of observations, normal distribution, and homogeneity of variances among the groups being compared. The Tukey's honestly significant difference (HSD) post hoc analysis results was used to determine pairwise differences between different treatment conditions. Subsequently, a Pearson's correlation analysis was conducted separately for each soil level. Finally, a Principal Component Analysis (PCA) was performed to identify relationships between variables and different treatments. All computations were carried out at a significance level set at $\alpha = 0.05$.

## 3. Results and Discussion

### 3.1. Changes in Soil pH, Sorption Capacity, and Carbonates as a Result of Soil Management Practices

ANOVA results (Table 1) show that for most variables the p-values for the main effects of year, treatment, and depth are generally highly significant ($p < 0.001$ or $p < 0.01$), indicating that these factors have a significant effect on these variables. The interactions (year × treatment, year × depth, treatment × depth) show varying levels of significance. For some variables, these interactions are not statistically significant ($p > 0.05$), suggesting that the effects of the factors may be independent.

**Table 1.** *p*-Values of a three-way analysis of variance (ANOVA) to assess the overall variability in the dependent variables attributed to treatment, sampling depth, and the year at $\alpha = 0.05$.

| | Variable | Year | Treatment | Depth | Year × Treatment | Year × Depth | Treatment × Depth |
|---|---|---|---|---|---|---|---|
| Total | C | 0.001 | <0.001 | <0.001 | 0.340 | 0.382 | 0.078 |
| | N | <0.001 | 0.006 | <0.001 | 0.618 | 0.580 | 0.039 |
| | S | <0.001 | 0.007 | <0.001 | 0.001 | 0.990 | 0.263 |
| | P | 0.012 | <0.001 | <0.001 | 0.303 | 0.269 | 0.157 |
| | K | <0.001 | <0.001 | 0.002 | 0.063 | 0.915 | 0.583 |
| | Ca | 0.474 | <0.001 | 0.016 | 0.045 | 0.710 | 0.021 |
| Available | P | 0.014 | <0.001 | <0.001 | 0.239 | 0.787 | 0.133 |
| | K | <0.001 | <0.001 | <0.001 | 0.105 | 0.053 | 0.001 |
| | Ca | <0.001 | <0.001 | 0.001 | 0.084 | 0.422 | 0.051 |
| | Cr | <0.001 | <0.001 | 0.001 | 0.043 | 0.523 | 0.287 |
| | Cu | 0.004 | 0.325 | <0.001 | 0.680 | 0.189 | 0.007 |
| | Ni | <0.001 | <0.001 | <0.001 | 0.042 | 0.291 | 0.099 |
| | Pb | <0.001 | <0.001 | <0.001 | 0.356 | 0.711 | 0.978 |
| | Zn | 0.004 | <0.001 | <0.001 | 0.553 | 0.077 | 0.064 |
| | Ha | <0.001 | 0.008 | 0.003 | 0.299 | 0.494 | 0.017 |
| | SBC | 0.001 | <0.001 | <0.001 | 0.002 | 0.490 | <0.001 |
| | CEC | 0.001 | <0.001 | <0.001 | 0.002 | 0.497 | <0.001 |
| | Bs (%) | 0.015 | 0.002 | 0.005 | 0.307 | 0.429 | 0.006 |
| | CaCO$_3$ | 0.088 | <0.001 | <0.001 | 0.005 | 0.020 | 0.400 |
| | pH | <0.001 | <0.001 | <0.001 | 0.003 | 0.858 | <0.001 |

Ha—hydrolytic acidity, SBC—sum of basic cations, CEC—cation exchange capacity, Bs—base saturation.

Soil pH is significantly influenced by the chemical composition of the bedrock and parent materials, as well as soil management practices [19]. On average, the pH values in the vineyard soil ranged from 6.7 to 7.7 (slightly acidic to slightly alkaline) for the 0–30 cm layer and from 7.5 to 7.9 (slightly alkaline) for the 30–60 cm depth. Limestone and dolomite, which are the bedrock in the vineyards in this study, release basic cations such as calcium and magnesium into the soil during weathering. These basic cations have good solubility and are naturally present in sufficient quantities due to their origin in the

parent material. As a result, they can counteract soil acidification caused by lower rates of NPK fertilizers and root secretions in the G + NPK1 and G treatments (Table 2). The application of poultry manure in a 4-year cycle in the P + FYM treatment had no significant effect on changes in soil pH. According to Busari et al. [20], poultry droppings have an alkaline pH, so their addition to alkaline soils does not fundamentally change the soil pH. However, the use of high doses of NPK (G + NPK2) disrupted the soil's buffering mechanism [21], leading to a statistically significant change in soil pH. High concentrations of nutrients, particularly $Ca^{2+}$, increased the solubility of basic cations while limiting the solubility of acidic cations [22] resulting in increased acidification in the G + NPK2 treatment. Higher soil pH enhances the electrostatic bonds between soil particles and exchangeable cations, thereby increasing the total sorption capacity [23]. This is supported by the findings in this study (Table 2). In the 0–30 cm layer, a higher dose of NPK significantly increased Ha but significantly decreased other soil sorption parameters compared to the other treatments. Sorption parameters are significantly influenced by soil texture and organic matter content [21,24]. However, in this study, carbonates played a more important role, and their dissolution was more pronounced in the T treatment, likely due to more intensive soil tillage. Soil tillage promotes aeration, supports mineralization processes, and enhances nutrient solubility [25]. This is reflected in the soil sorption capacity, with lower values of SBC and CEC observed in the T treatment compared to P + FYM and G, especially in the 30 cm layer (Table 2). As the NPK dose increased, there was a more intensive decrease in basic cations (SBC) and a reduction in CEC due to the limited solubility of acidic cations [26]. This effect was more pronounced in the 0–30 layer, indicating a greater impact of human interventions such as tillage and fertilization. Compared to the T treatment, the soil sorption capacity in the 30–60 cm layer significantly improved in the following order: G + NPK2 < P + FYM < G + NPK1 < G. Although Bs values were statistically significantly reduced with the application of higher NPK doses (G + NPK2) compared to other soil management practices in both soil layers, it is noteworthy that the values were ≤95 (Table 2), indicating that the soil sorption complex remained fully saturated [21].

**Table 2.** Means and standard deviations for all studied variables attributed to soil management practices in vineyards. For each variable, the same letters in rows indicate homogenous groups of treatments, as determined by Tukey's analysis at α = 0.05.

| | Grass Strips | Tillage | Plowed FYM | Grass Strips + NPK 1 | Grass Strips + NPK 2 |
|---|---|---|---|---|---|
| **0–30 cm** | | | | | |
| Ha (mmol kg$^{-1}$) | 3.9 ± 1.4 [a] | 3.9 ± 1.7 [a] | 4.6 ± 1.7 [ab] | 5.4 ± 1.9 [ab] | 10.6 ± 7.7 [b] |
| SBC (mmol kg$^{-1}$) | 470.9 ± 5.8 [d] | 303.9 ± 25.9 [ab] | 431.6 ± 14.6 [cd] | 363.0 ± 53.7 [bc] | 247.1 ± 97.2 [a] |
| CEC (mmol kg$^{-1}$) | 474.8 ± 6.5 [d] | 307.8 ± 24.6 [ab] | 436.2 ± 14.5 [cd] | 368.3 ± 52.2 [bc] | 257.7 ± 91.3 [a] |
| Bs (%) | 99.2 ± 0.3 [b] | 98.7 ± 0.6 [b] | 99.0 ± 0.4 [b] | 98.5 ± 0.7 [b] | 94.9 ± 4.1 [a] |
| CaCO$_3$ (g kg$^{-1}$) | 2.0 ± 0.1 [b] | 0.6 ± 0.1 [a] | 1.4 ± 0.2 [ab] | 1.3 ± 0.5 [ab] | 0.8 ± 0.9 [a] |
| pH | 7.7 ± 0.3 [b] | 7.6 ± 0.3 [b] | 7.7 ± 0.3 [b] | 7.3 ± 0.4 [b] | 6.7 ± 0.4 [a] |
| **30–60 cm** | | | | | |
| Ha (mmol kg$^{-1}$) | 3.3 ± 1.3 [a] | 3.4 ± 1.6 [a] | 4.0 ± 1.8 [a] | 3.5 ± 1.5 [a] | 3.8 ± 1.9 [a] |
| SBC (mmol kg$^{-1}$) | 489.5 ± 4.7 [c] | 295.8 ± 24.0 [a] | 434.2 ± 38.8 [bc] | 459.5 ± 24.6 [c] | 373.9 ± 59.2 [b] |
| CEC (mmol kg$^{-1}$) | 492.8 ± 5.3 [c] | 299.2 ± 22.7 [a] | 438.1 ± 39.1 [bc] | 462.9 ± 25.0 [c] | 377.7 ± 57.5 [b] |
| Bs (%) | 99.3 ± 0.3 [b] | 98.9 ± 0.6 [a] | 99.1 ± 0.4 [b] | 99.3 ± 0.3 [b] | 98.9 ± 0.7 [a] |
| CaCO$_3$ (g kg$^{-1}$) | 2.6 ± 0.7 [b] | 0.8 ± 0.1 [a] | 1.9 ± 1.1 [ab] | 2.1 ± 0.5 [b] | 1.4 ± 0.5 [ab] |
| pH | 7.9 ± 0.3 [b] | 7.8 ± 0.3 [b] | 7.8 ± 0.3 [b] | 7.8 ± 0.3 [b] | 7.5 ± 0.2 [a] |

Ha—hydrolytic acidity, SBC—sum of basic cations, CEC—cation exchange capacity, Bs—base saturation.

*3.2. Changes in Total and Available Nutrients as a Result of Soil Management Practices*

The lowest total C content in both layers was observed in the T treatment and it increased in the following order: G + NPK1 < G + NPK2 < P + FYM < G for the 0–30 cm layer and G + NPK2 < G + NPK1 < G < P + FYM for the 30–60 cm layer. Aeration and subsequent soil organic matter oxidation result from soil tillage [27], leading to the stabilization of total C content, which was lowest in this treatment. On the other hand, the application of organic matter/manures/additives to the soil [1] as well as the creation of conditions for increased biomass production through higher available nutrient content [28], can be the result of a higher C content, as also evidenced by the findings in this study (Table 3). Total N content ranged from 1467 to 1936 mg kg$^{-1}$, in the first soil layer (0–30 cm) and from 1029 to 1254 mg kg$^{-1}$ in the second soil layer (30–60 cm). Changes in total N content were influenced by the soil management in the vineyard (Table 3). In the P + FYM, G, and G + NPK2 treatments, total N content significantly increased by 429, 386, and 469 mg kg$^{-1}$, respectively, compared to the T treatment in the 0–30 cm layer. This increase in N content can be attributed to the addition of organic matter from poultry droppings and indirectly from the created grass biomass, supported by mineral fertilization. This resulted in higher SOM content, especially its stable fraction, which is more resistant to microbial decomposition [29]. Total S content fluctuated in the range of 164–363 mg kg$^{-1}$ depending on the soil management practices in the vineyard, as well as the soil layer, but without statistical significance. The total P content in the soil depends on the level and intensity of P fertilization [29]. While the P content in FYM is not high and depends on several factors [16], the content of P in poultry manure is 2 to 4 times higher than in other manures and ranges from 13.6 to 25.4 g P$_2$O$_5$ kg$^{-1}$ dm [30]. Poultry manure applied in P + FYM treatment in this study contained 1.3% P$_2$O$_5$, and its application in a 4-year cycle had a statistically significant effect on increasing P levels in the soil compared to the T treatment (Table 3) in the 0–30 cm soil layer.

The annual application of 50 kg ha$^{-1}$ P in the G + NPK2 treatment significantly increased total P levels in both soil layers compared to the T treatment. This effect was not observed in the case of a lower dose of NPK (G + NPK1). These results indicate the accumulation of total P, primarily due to the supply of P from poultry manure but also from the annual application of high rates of NPK. The accumulation of P in the soil depends on factors such as clay content, mineralogical composition, organic matter content, and soil pH [31]. In the case of alkaline soils (as observed in this study), the increase in total P content occurs through chemosorption [32]. However, with high rates of NPK application, soil acidification occurred (as indicated by the lowest pH—see Table 2), which enhanced the availability of P compared to the other treatments. The availability of P from its total content was relatively high, if we consider the fact that the overall supply of P in the soils of Slovakia (compared to other elements such as K, Ca, Mg) is lower and ranges from 0.03–0.2%. However, only 1–8% of the total P supply can be used by plants to produce yields [10,16,29]. On average, with a harvest of 10 t of grapes ha$^{-1}$, the vine takes 60–80 kg of P ha$^{-1}$ from the soil. In G + NPK2, G + NPK1, P + FYM, T and G, the contents of available P from the total supply represented 41, 31, 30, 29, and 22%, respectively. Vine roots can grow to a depth of more than 1 m [33], so information about nutrient availability in the 30–60 cm layer has its justification (Table 3). The highest availability of P from total P was found in GNPK2 (32%) > P + FYM (26%) > G + NPK1 (21%) > T (19%) = G (19%). In the 0–30 cm layer, the contents of available P in all management practices were above 200 mg kg$^{-1}$, which is considered very high for sandy loam and loamy soils and often does not require additional P fertilization (following criteria for evaluating available P content in arable soil—Mehlich III. [29]). Even in the 30–60 cm layer, the content of available P exceeded this limit value of 200 mg kg$^{-1}$. Grassy strips (G) and the application of both NPK rates (G + NPK1, G + NPK2) increased the content of total K in both layers in comparison with T and P + FYM treatments. This occurs primarily because of physicochemical and biological sorption and to a lesser extent fixation [10]. However, the improvement in K availability was predominantly observed in the top layer (Table 3). Notably, poultry

droppings contained 1.2% $K_2O$, which did not significantly affect its total supply but increased its availability. The content of available K shifted from a satisfactory supply in the T treatment to a good supply in the P + FYM treatment. In other treatments, the content of available K also improved to a good supply, and in the case of G + NPK2, it even reached a high supply, suggesting that K fertilization could be omitted for several years [29]. Vineyard soil primarily consists of limestone and dolomite [12], so it is not surprising that the total Ca contents ranged from 6.3 to 15.1% (Table 2). Soil management practices, including tillage, aeration, mixing, and fertilization, influence changes in both the total Ca stock and its availability [10,29]. The highest total Ca contents were found after the application of poultry manure in the P + FYM treatment and the grassy strips in the G treatment in both layers. Additionally, in the second layer (30–60 cm), the G treatment had the highest Ca content. Due to the suction of water from the soil, the roots of grass create conditions for the precipitation of carbonates, which is also documented by the highest $CaCO_3$ values in the G treatment (Table 2). On the other hand, root exudates can increase the solubility of $CaCO_3$ and, consequently, Ca leaching or making it available to plants (Table 3). High doses of NPK, however, had a negative effect on the content of available Ca compared to other treatments in the 0–30 cm layer. In the 30–60 cm layer, the highest content of available Ca was found in G and gradually decreased in the following order: G + NPK1 > P + FYM > G + NPK2 > T. However, the availability of Ca from its total stock was slightly different. In 0–30 cm and in 30–60 cm, it ranged from 49–65% and 53–71%, depending on soil management practices. At both depths, tillage, and application of NPK fertilizers had the most significant effect on Ca availability from total Ca.

**Table 3.** Means and standard deviations for all studied variables attributed to soil management practices in vineyards. For each variable, the same letters in rows indicate homogenous groups of treatments, as determined by Tukey's analysis at $\alpha$ = 0.05.

| | | | **Grass Strips** | **Tillage** | **Plowed FYM** | **Grass Strips + NPK 1** | **Grass Strips + NPK 2** |
|---|---|---|---|---|---|---|---|
| | | | 0–30 cm | | | | |
| Total | C | g kg$^{-1}$ | 19.5 ± 4.19 [b] | 12.4 ± 0.34 [a] | 18.8 ± 1.51 [b] | 17.1 ± 2.36 [b] | 17.8 ± 3.42 [b] |
| | N | mg kg$^{-1}$ | 1853.6 ± 330.6 [b] | 1467.4 ± 99.7 [a] | 1896.1 ± 136.3 [b] | 1709.6 ± 168.9 [ab] | 1936.3 ± 385.0 [b] |
| | S | mg kg$^{-1}$ | 248.9 ± 56.1 [a] | 244.5 ± 70.2 [a] | 278.5 ± 56.4 [a] | 251.4 ± 101.2 [a] | 362.5 ± 215.9 [a] |
| | P | mg kg$^{-1}$ | 909.2 ± 126.9 [ab] | 772.8 ± 106.8 [a] | 1216.3 ± 134.6 [bc] | 1113.8 ± 136.2 [abc] | 1376.4 ± 434.3 [c] |
| | K | g kg$^{-1}$ | 25.0 ± 5.41 [b] | 19.5 ± 3.52 [a] | 18.1 ± 1.67 [a] | 26.1 ± 2.68 [b] | 29.8 ± 4.51 [c] |
| | Ca | g kg$^{-1}$ | 12.8 ± 1.59 [b] | 7.86 ± 1.10 [a] | 12.5 ± 2.01 [b] | 8.28 ± 1.76 [a] | 6.34 ± 1.64 [a] |
| Available | P | mg kg$^{-1}$ | 201.0 ± 98.7 [a] | 222.0 ± 104.5 [a] | 369.2 ± 87.7 [ab] | 350.0 ± 127.7 [ab] | 560.3 ± 248.9 [b] |
| | K | mg kg$^{-1}$ | 289.7 ± 83.9 [ab] | 208.1 ± 40.2 [a] | 350.1 ± 106.9 [bc] | 390.3 ± 108.4 [bc] | 453.5 ± 133.7 [c] |
| | Ca | g kg$^{-1}$ | 6.77 ± 1.51 [c] | 5.12 ± 0.99 [b] | 6.14 ± 0.99 [bc] | 5.22 ± 1.02 [b] | 3.52 ± 0.47 [a] |
| | | | 30–60 cm | | | | |
| Total | C | g kg$^{-1}$ | 12.1 ± 2.19 [bc] | 8.77 ± 1.15 [a] | 12.4 ± 2.31 [c] | 10.7 ± 1.25 [abc] | 9.59 ± 0.92 [ab] |
| | N | mg kg$^{-1}$ | 1 077.8 ± 133.4 [ab] | 1 081.8 ± 147.4 [ab] | 1 254.9 ± 272.0 [b] | 1 027.8 ± 115.9 [a] | 1 062.7 ± 128.2 [ab] |
| | S | mg kg$^{-1}$ | 183.4 ± 80.7 [a] | 189.7 ± 44.1 [a] | 210.6 ± 70.3 [a] | 164.0 ± 69.0 [a] | 218.4 ± 145.0 [a] |
| | P | mg kg$^{-1}$ | 640 ± 90 [ab] | 490 ± 100 [a] | 730 ± 240 [ab] | 600 ± 110 [ab] | 780 ± 290 [b] |
| | K | g kg$^{-1}$ | 27.0 ± 3.97 [b] | 20.5 ± 3.62 [a] | 20.1 ± 3.38 [a] | 28.2 ± 3.97 [b] | 30.1 ± 4.54 [b] |
| | Ca | g kg$^{-1}$ | 15.1 ± 1.45 [c] | 6.82 ± 0.62 [a] | 12.2 ± 3.27 [bc] | 11.4 ± 1.18 [b] | 7.65 ± 1.15 [a] |
| Available | P | mg kg$^{-1}$ | 122.8 ± 43.5 [a] | 93.9 ± 59.3 [a] | 188.3 ± 95.9 [ab] | 127.6 ± 43.3 [a] | 248.5 ± 136.1 [b] |
| | K | mg kg$^{-1}$ | 182.3 ± 56.2 [a] | 141.6 ± 19.9 [a] | 178.4 ± 91.7 [a] | 170.5 ± 39.3 [a] | 200.7 ± 32.9 [a] |
| | Ca | g kg$^{-1}$ | 7.99 ± 2.75 [b] | 4.82 ± 0.75 [a] | 6.41 ± 1.20 [ab] | 6.49 ± 0.90 [ab] | 5.36 ± 0.40 [a] |

### 3.3. Changes in Risk Elements as a Result of Soil Management Practices

Monitoring the presence of hazardous elements in soils is of utmost importance, as changes in soil pH, for instance, can enhance their mobility, leading to potential transfer into plants and, subsequently, the entire food chain [8,34,35]. In the Slovak Republic, the permissible limit for hazardous elements in soils with different soil textures is regulated by Act 220/2004 [36]. According to these regulations, the concentrations of Cr, Cu, Ni, Pb, and Zn should not exceed 70, 60, 50, 70, and 150 mg kg$^{-1}$, respectively, for sandy loam and loamy soils. In case of this study, before the experiment, the soil 56.9% sand, 33% silt, and 10.1% clay on average and it is classified as sandy loam in texture according to IUSS Working Group WRB [14]. However, only Cu content significantly exceeded the specified limits. Moreover, these excess levels were significantly higher in the 0–30 cm layer compared to the 30–60 cm soil layer (Figure 2). Specifically, in the 0–30 cm layer, Cu concentrations in the T, P + FYM, G, G + NPK1, and G + NPK2 treatments exceeded the limits by 62.6, 73.7, 70.2, 82.1, and 102.9 mg kg$^{-1}$, respectively, following Slovak Republic regulations. Cu generally shows low mobility in soils and tends to accumulate in the first few centimeters due to its tendency to form organometallic complexes with organic matter [37] and for this reason the results are not surprising. In the 30–60 cm layer, Cu concentrations slightly exceeded the limits (by 5, 0.2, and 4.6 mg kg$^{-1}$, respectively) only in the P + FYM, G, and G + NPK1 treatments. Historically, there was a vineyard in this area that was intensively tilled, so it is possible that due to soil tillage, the concentration of Cu also increased in the lower soil layer. As reported by Kabata-Pendias [37], Cu can be bonded to carbonates, clay minerals, and Mn and Fe oxyhydroxides. This makes it possible that Cu can be found in deeper soil horizons. The issue of elevated Cu content primarily affects vineyard soils in Slovakia and can be attributed to the historical use of Cu-based fungicides [38]. Similar problems have been observed in vineyard soils worldwide [39,40]. Over more than 150 years of Cu-based fungicide use, these elements accumulated significantly in vineyard soils, leading to soil degradation and environmental challenges in various ecosystem components [40]. The concentrations of other hazardous elements remained within acceptable limits, and their distribution in both soil layers was influenced by the specific vineyard soil management practices (Figure 2). Hazardous elements generally exhibit lower mobility and tend to accumulate in the upper layer of soils, often forming organic bonds with organic substances [37]. Consequently, it is not surprising that higher concentrations were observed in the 0–30 cm layer compared to the 30–60 cm. Notably, the application of poultry manure in the P + FYM treatment had the most significant effect, increasing the content of Pb and Zn, but also Cr, Cu, Ni, Pb, and Zn in both the 0–30 cm and 30–60 cm layers compared to other soil management practices in the vineyard. Particularly, the application of higher rates of NPK (G + NPK2) resulted in the lowest contents of Cr, Ni, and Pb in the 0–30 cm layer as well as the lowest concentrations of all hazardous elements in the 30–60 cm layer. The content of Zn was significantly below the threshold limit in all treatments, even though Dithane M45 based on the active substance Mancozeb containing Zn was used for spraying against fungal diseases. In the places where the vineyard is located, no geochemical anomaly is recorded [41], so the content of risk elements in the soil is influenced by human activities through fertilization and plant protection. In addition, fertilizer producers do not indicate the content of harmful elements on the labels, but during registration of fertilizers, this issue is strictly monitored by the control authority in the Slovak Republic. Such a practice eliminates a significant increase in harmful elements in soil, including vineyards.

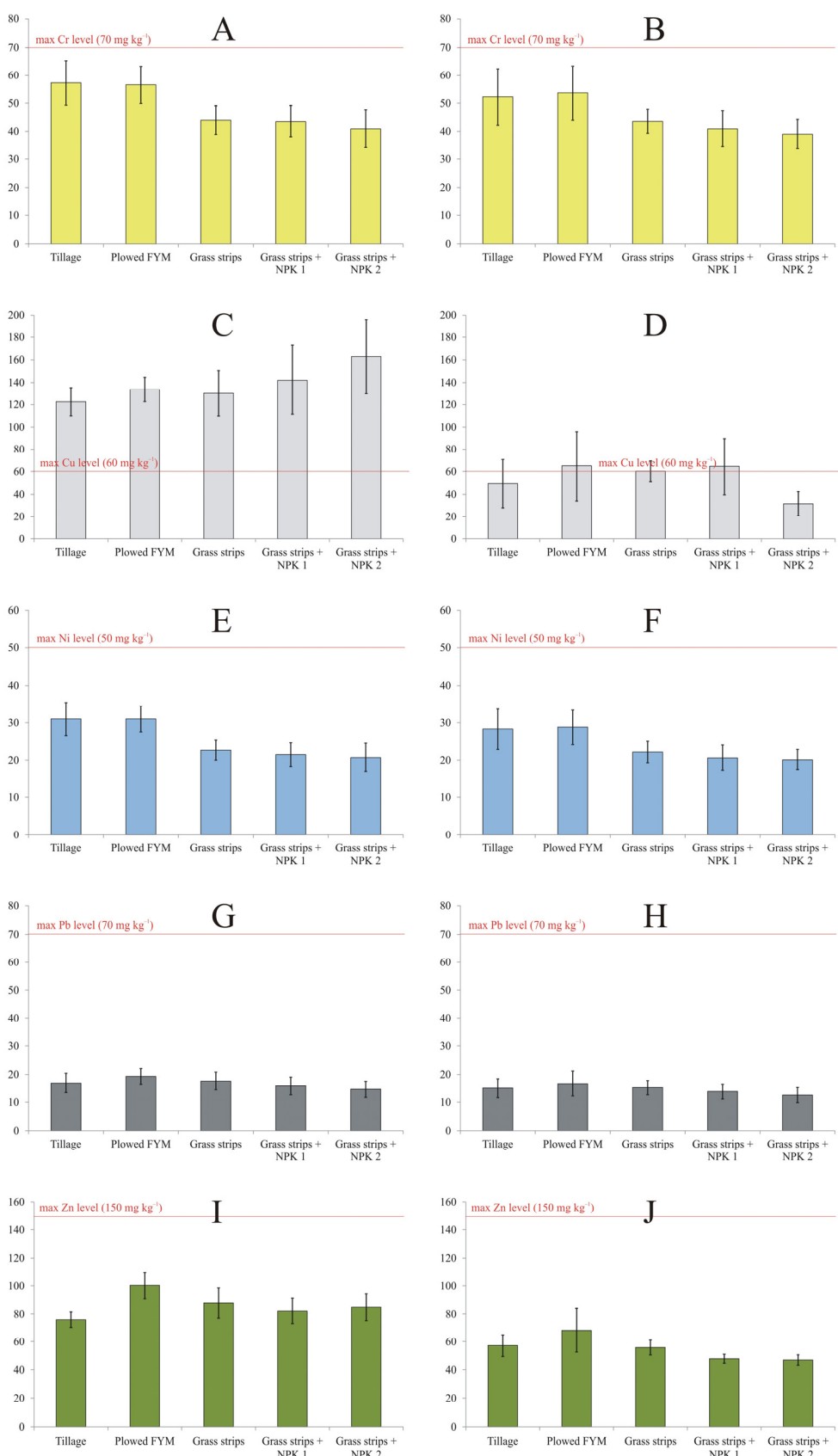

**Figure 2.** Means and standard deviations for all studied variables attributed to soil management practices in vineyards, for 0–30 cm (**A**,**C**,**E**,**G**,**I**) and 30–60 cm (**B**,**D**,**F**,**H**,**J**) layers. The red line means the threshold limit allowed by Act No. 220/2004 and Decree No. 59/2013.

### 3.4. Relationships between Soil Properties in the Soil of the Vineyard

The impact of different soil management practices in the vineyard on soil properties and their relationships based on PCA is depicted in Figure 3. Principal components PC1 and PC2 collectively explain approximately 77% of the total variance in the dataset. Variables such as total C, N, P, and S total, as well as Cu, Zn, and Ha, are correlated, with their elevated values consistently observed in the 0–30 cm layer across most treatments (T, P + FYM, G + NPK1, and G + NPK2), except for the grass strips. Notably, P + FYM exhibits significantly high values for Pb, Ni, and Cr, while a high C total is observed in both P + FYM and G + NPK1 treatments. Total and available Ca, soil pH, $CaCO_3$, SBC, CEC, and Bs are also positively correlated, with their peak values consistently recorded in the 30–60 cm layer across all treatments, reaching their highest levels in the grass strip (G) and G + NPK1 treatments. On the other hand, total K demonstrates its highest values in the case of G + NPK2 in the second soil layer (30–60 cm). Treatments involving NPK show negative values for Pb, Ni, and Cr. In the biplot, treatments with similar positions are more alike in terms of the studied variables.

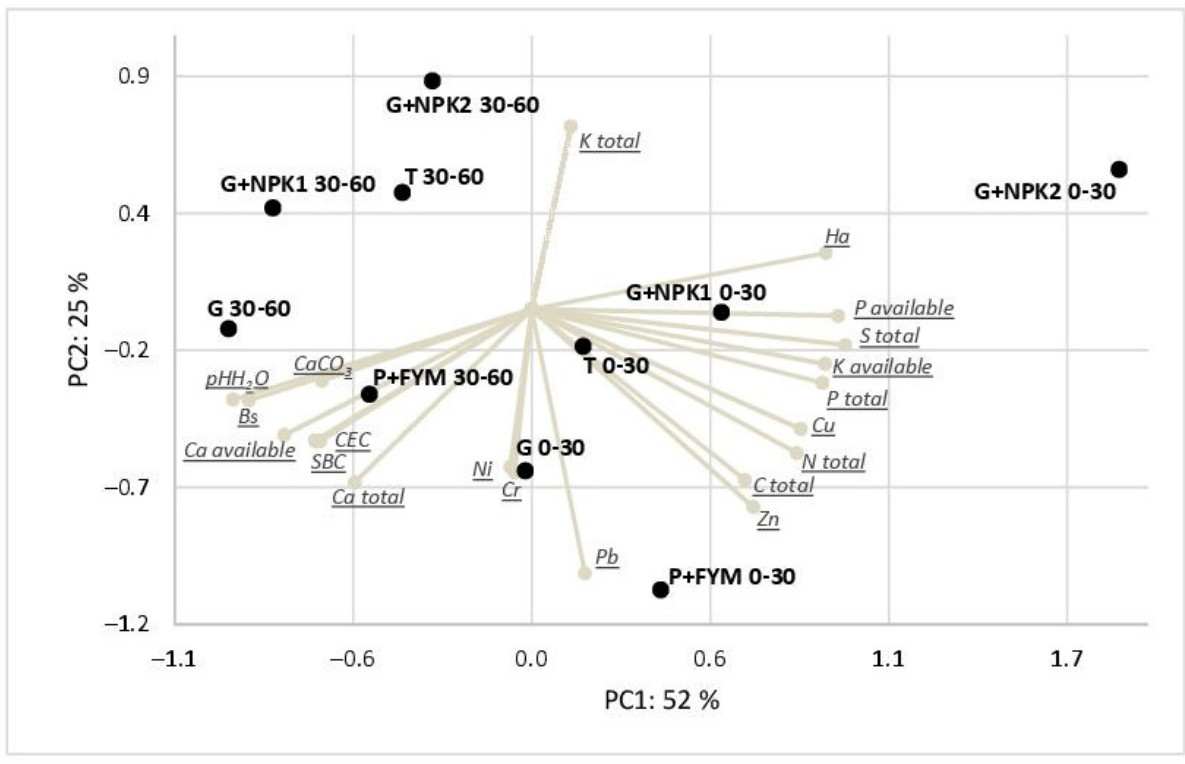

**Figure 3.** Visual representation of the different treatment conditions in the reduced-dimensional PCA space defined by PC1 and PC2.

In the 0–30 layer, more intensive interventions due to soil management practices were observed compared to the second layer (30–60 cm), and this was also reflected in the statistical significance and strength of relationships between soil properties in the vineyard (Table 4). In the 0–30 cm soil layer, a very strong positive correlation (r = 0.89) was found between total C and total N, indicating that as carbon content increases, nitrogen content also significantly rises. This relationship is unsurprising, as organic matter is the primary source of both C and N in soil [42,43]. Additionally, C can exist in soil, often in the form of carbonates [44]. Given that the vineyard's bedrock consisted of dolomite and limestone [12], the $CaCO_3$ content in the vineyard soil ranges from 0.6 to 2.8%, depending on the soil management practices (Table 2). Notably, the carbonate content was highest in the G treatment but decreased due to dissolution during soil acidification resulting from the application of high rates of NPK fertilizer. In the 0–30 cm layer, a positive correlation was

recorded between total C and $CaCO_3$, indicating that carbonate content directly influenced the increase in total C in the vineyard soil. A moderate positive correlation (r = 0.42) exists between total C and total P, suggesting a positive but less pronounced relationship between carbon and phosphorus content. Negative correlations were found between total and available P and $CaCO_3$, as well as total and available Ca, indicating that chemosorption is not the primary mechanism responsible for increasing soil P. This implies that the increase in the content of total P in the soil is likely due to its incorporation and stabilization in the organic matter of the soil [31]. A moderate negative correlation (r = −0.47) between total C and Cr implies that as the carbon content increases, Cr concentration in the aqueous solution tends to decrease. A similar correlation (r = −0.46) is observed between total C and Ni. More mobile and soluble fractions of soil organic matter play a significant role in immobilizing hazardous elements by forming organo-metallic complexes [45]. Interesting positive correlations were found between total C and Zn, as well as $CaCO_3$ and Zn (Table 4). No significant correlation was observed between soil pH and Zn. According to Kabata-Pendias [37] and Kováčik and Ryant [29], soil pH typically influences the mobility of Zn in soil. When Zn content is low, incorporating organic matter into the soil supports the mobility of Zn in the soil. However, in cases of increased Zn supply in the soil, as observed in vineyards with long-term Zn-based fungicide applications [39,40], this effect becomes negative [29]. In the 30–60 cm soil layer, a moderately strong positive correlation was found between total C total and S (r = 0.63), indicating that they tend to increase together. Within C total, all significant correlations are positive, including Cu (r = 0.64), Zn (r = 0.54), SBC (r = 0.63), CEC (r = 0.63), and $CaCO_3$ (r = 0.42).

**Table 4.** Pearson's correlation coefficients for two soil layers: 0–30 (left upper triangle) and 30–60 (right bottom triangle). All significant correlations are marked in red font. Negative correlations are indicated by blue cells.

| 0–30 (upper) / 30–60 (lower) | | C | N | S | P | K | Ca | P | K | Ca | Cr | Cu | Ni | Pb | Zn | Ha | SBC | CEC | Bs | CaCO3 | pH |
|---|---|---|---|---|---|---|---|---|---|---|---|---|---|---|---|---|---|---|---|---|---|
| | | Total | | | | | | Available | | | | | | | | | | | | | |
| Total | C | | 0.89 | 0.23 | 0.42 | 0.06 | 0.34 | 0.14 | 0.14 | −0.05 | −0.47 | −0.09 | −0.46 | −0.21 | 0.50 | 0.25 | 0.36 | 0.38 | −0.15 | 0.45 | −0.15 |
| | N | 0.73 | | 0.27 | 0.55 | 0.06 | 0.17 | 0.31 | 0.12 | −0.26 | −0.42 | 0.02 | −0.40 | −0.26 | 0.33 | 0.46 | 0.12 | 0.14 | −0.37 | 0.17 | −0.22 |
| | S | 0.18 | 0.04 | | 0.44 | 0.42 | −0.20 | 0.33 | −0.05 | −0.28 | −0.06 | 0.42 | −0.07 | 0.03 | 0.26 | −0.15 | 0.01 | 0.01 | 0.12 | −0.07 | −0.38 |
| | P | 0.35 | 0.34 | 0.52 | | 0.44 | −0.15 | 0.90 | 0.50 | −0.25 | −0.16 | 0.38 | −0.19 | −0.03 | 0.26 | 0.50 | −0.23 | −0.22 | −0.46 | −0.18 | −0.49 |
| | K | 0.02 | −0.18 | 0.34 | 0.52 | | −0.38 | 0.48 | 0.25 | −0.04 | −0.29 | 0.70 | −0.36 | 0.08 | 0.00 | 0.16 | −0.16 | −0.16 | −0.20 | −0.01 | −0.38 |
| | Ca | 0.64 | 0.15 | −0.09 | 0.06 | 0.08 | | −0.41 | −0.27 | 0.67 | 0.16 | 0.35 | 0.49 | −0.42 | | | 0.83 | 0.84 | 0.51 | 0.69 | 0.62 |
| Available | P | 0.06 | 0.12 | 0.41 | 0.92 | 0.51 | −0.14 | | 0.64 | −0.32 | −0.05 | 0.42 | −0.09 | 0.00 | 0.04 | 0.65 | −0.52 | −0.51 | −0.65 | −0.48 | −0.62 |
| | K | −0.23 | −0.35 | −0.25 | 0.26 | 0.17 | 0.09 | 0.46 | | −0.28 | −0.39 | 0.18 | −0.36 | −0.24 | −0.03 | 0.60 | −0.45 | −0.43 | −0.61 | −0.29 | −0.65 |
| | Ca | 0.18 | −0.18 | −0.12 | 0.14 | 0.33 | 0.63 | 0.16 | 0.47 | | 0.45 | −0.05 | 0.47 | 0.70 | 0.32 | −0.53 | 0.71 | 0.71 | 0.58 | 0.53 | 0.72 |
| | Cr | 0.08 | 0.30 | 0.14 | 0.35 | −0.22 | −0.07 | 0.34 | 0.14 | 0.15 | | 0.10 | 0.97 | 0.77 | 0.19 | −0.34 | 0.13 | 0.12 | 0.35 | −0.22 | 0.46 |
| | Cu | 0.64 | 0.57 | −0.03 | 0.32 | 0.00 | 0.37 | 0.13 | −0.01 | 0.30 | 0.43 | | 0.06 | 0.41 | 0.11 | 0.07 | −0.16 | −0.16 | −0.15 | −0.20 | −0.21 |
| | Ni | 0.02 | 0.31 | 0.10 | 0.27 | −0.28 | −0.08 | 0.28 | 0.08 | 0.11 | 0.98 | 0.35 | | 0.75 | 0.16 | −0.37 | 0.13 | 0.12 | 0.35 | −0.22 | 0.51 |
| | Pb | 0.19 | 0.22 | 0.21 | 0.53 | 0.15 | 0.18 | 0.50 | 0.25 | 0.47 | 0.88 | 0.48 | 0.83 | | 0.39 | −0.38 | 0.40 | 0.39 | 0.38 | 0.10 | 0.49 |
| | Zn | 0.54 | 0.63 | 0.10 | 0.41 | −0.34 | 0.18 | 0.28 | 0.12 | 0.16 | 0.78 | 0.69 | 0.73 | 0.68 | | −0.18 | 0.50 | 0.51 | 0.25 | 0.46 | 0.15 |
| | Ha | 0.05 | 0.19 | −0.53 | 0.02 | −0.32 | −0.09 | 0.10 | 0.31 | 0.00 | 0.05 | 0.08 | 0.00 | −0.09 | 0.25 | | −0.66 | −0.63 | −0.97 | −0.49 | −0.66 |
| | SBC | 0.63 | 0.09 | 0.06 | 0.22 | 0.32 | 0.84 | 0.02 | 0.08 | 0.57 | −0.18 | 0.42 | −0.24 | 0.14 | 0.09 | −0.15 | | 1.00 | 0.76 | 0.81 | 0.66 |
| | CEC | 0.63 | 0.09 | 0.05 | 0.22 | 0.32 | 0.84 | 0.02 | 0.08 | 0.57 | −0.18 | 0.42 | −0.24 | 0.14 | 0.10 | −0.13 | 1.00 | | 0.74 | 0.81 | 0.65 |
| | Bs | 0.21 | −0.11 | 0.46 | 0.09 | 0.37 | 0.41 | −0.05 | −0.20 | 0.23 | −0.11 | 0.15 | −0.10 | 0.14 | −0.13 | −0.88 | 0.56 | 0.55 | | 0.57 | 0.69 |
| | CaCO3 | 0.42 | 0.08 | −0.18 | 0.03 | 0.26 | 0.79 | −0.12 | −0.05 | 0.61 | −0.19 | 0.33 | −0.17 | 0.06 | −0.05 | −0.14 | 0.83 | 0.83 | 0.45 | | 0.51 |
| | pH | 0.25 | 0.35 | −0.16 | −0.13 | −0.01 | 0.37 | −0.27 | −0.29 | 0.23 | 0.28 | 0.46 | 0.36 | 0.34 | 0.21 | −0.48 | 0.28 | 0.27 | 0.52 | 0.50 | |

The results of this study are primarily limited by the current soil and climate conditions of the studied region. Our findings can be applied to temperate climate regions with sandy loam Rendzic Leptosols developed on limestone or dolomite or areas that are characterized by similar soil properties. Overall, the data confirmed the significant effect of soil management practices in a productive vineyard on soil pH, sorption capacity, nutrient regime, and the content of the hazardous elements in the soil (Figure 3). Since the data on the representation of hazardous elements in the vine organs were not determined, it is difficult to predict the transfer of elements from the soil into the plant and subsequently into the food chain. However, information about the content of nutrients and hazardous substances in the soil has its justification, because helps to predict future risks. For example, the mobility of nutrients [31,32] and hazardous elements [37,45] depends on soil pH changes, clay content and soil organic matter content. There is a potential risk of Cu mobility as a result of long-term fungicide application [38] after soil pH changes and SOM reduction as a result of intensive soil management [3,19,46].

## 4. Conclusions

The higher level of NPK fertilization in the grass strips between vine rows resulted in the highest reserves of total N, P, K, and available P and K, it significantly diminished the overall soil sorption capacity. High NPK level led to reduced soil pH, resulting in decreased total and available Ca stocks in the soil. Conversely, intensive soil tillage reduced the stocks of total C, N, P, and available K. Hazardous elements, such as Cr, Ni, Pb, and Zn, did not exceed the established limit values in any of the soil management practices within the vineyard, except for Cu. Cu content exceeded permissible levels, ranging from 62.6 mg kg$^{-1}$ (in T) to 102.9 mg kg$^{-1}$ (in G + NPK2), in compliance with Slovak Republic regulations.

The results of this study have their limitations, such as the specific soil and climate conditions and the tested soil management practices, but even so, from both ecological and economic standpoints, the most optimal approach for winegrowers in temperate climate zones, particularly in areas with similar soil conditions, involves the implementation of grass strips without NPK fertilization. Grass strips have proven to be an environmentally friendly alternative to conventional or intensive vine cultivation. They maintain a favorable sorption capacity and an optimal nutrient regime within the soil, despite minimal human intervention when compared to other soil management practices in vineyards.

**Author Contributions:** Conceptualization, V.Š.; methodology, V.Š.; software, E.W.-G.; validation, V.Š., J.J. and J.H.; formal analysis, V.Š., J.J. and J.H.; investigation, V.Š.; resources, V.Š. and J.H.; data curation, J.J. and E.W.-G.; writing—original draft preparation, V.Š.; writing—review and editing, V.Š., E.W.-G., J.J. and J.H.; visualization, J.J.; supervision, V.Š.; project administration, V.Š. and J.H.; funding acquisition, J.H. All authors have read and agreed to the published version of the manuscript.

**Funding:** This study was supported by the Scientific Grant Agency, grant number VEGA 1/0116/21 and VEGA 1/00201/22. Further, this publication is the result of the implementation of the projects by the Slovak Research and Development Agency under the contract No. APVV-21-0089.

**Data Availability Statement:** The datasets generated and analyzed during the current study are available from the authors upon a reasonable request.

**Acknowledgments:** The authors express their gratitude to the editor and the reviewers for their constructive comments.

**Conflicts of Interest:** The authors declare no conflict of interest. The funders had no role in the design of the study; in the collection, analyses, or the interpretation of the data; in the writing of the manuscript, or in the decision to publish the results.

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
