# Peer review of "Optimizing Soil Management for Sustainable Viticulture: Insights from a Rendzic Leptosol Vineyard in the Nitra Wine Region, Slovakia"

_agronomy, doi:10.3390/agronomy13123042_

Round 1

Reviewer 1 Report

Comments and Suggestions for Authors

This paper deals with the effect of diverse alternatives to the management of a vineyard on soil nutrient balance and pollutants related with treatments against fungal attacks.  Treatments are variated and include tillage modification, grass strips and combinations of organic and mineral fertilizers.

The subject addressed is important in the current crucial moment with agricultural management in a turning point towards environmental sustainability.

The results might be interesting, but the conceptual framework of the work is very week. The authors do not describe the treatments properly and the choice of these particular treatments and treatment levels is not justified.  The “control” management of the vineyard is not described in the text.

The spatial design of the experiment must be clearly explained.  

As presented in the tables, the statistical analysis is very poor. Their two-way ANOVA design is not appropriate for an experiment that includes three factors with several levels each (treatment -several levels-, sampling depth -two levels- and sampling date -3 levels-)

A consistent discussion of the results with existing knowledge in mind is also required.

Therefore, the paper needs to be improved substantially before being resubmitted for possible evaluation

Specific demands are:

-        Please, describe the spatial design of the experiment: Are you using a unique vineyard? If yes, there is pseudo replication in the design; how big is the vineyard or vineyards? Please clearly describe the spatial design of the experimental plots.

-        Please, describe correctly the “control” treatment, corresponding to the pre-experimental management of the vineyard.

-        Provide analyses of the organic materials used as manure.

-        The statistical treatment is not clear: (a) the tables do not include control values; (2) sampling time -which year- is not specified; (c) are all sampling levels pooled? 

-        Improve the introduction and create a discussion.

-        Review the text for some minor language mistakes.

Comments on the Quality of English Language

English is correct.

Only some minor errors must be corrected

Author Response

Reviewer 1

This paper deals with the effect of diverse alternatives to the management of a vineyard on soil nutrient balance and pollutants related with treatments against fungal attacks.  Treatments are variated and include tillage modification, grass strips and combinations of organic and mineral fertilizers. The subject addressed is important in the current crucial moment with agricultural management in a turning point towards environmental sustainability. The results might be interesting, but the conceptual framework of the work is very week. The authors do not describe the treatments properly and the choice of these particular treatments and treatment levels is not justified.  The “control” management of the vineyard is not described in the text. The spatial design of the experiment must be clearly explained. As presented in the tables, the statistical analysis is very poor. Their two-way ANOVA design is not appropriate for an experiment that includes three factors with several levels each (treatment -several levels-, sampling depth -two levels- and sampling date -3 levels-) A consistent discussion of the results with existing knowledge in mind is also required. Therefore, the paper needs to be improved substantially before being resubmitted for possible evaluation.

Response: Thank you very much for the constructive criticism and suggestions for improving the manuscript. We revised the manuscript and rewrote some parts. We have improved the description of the trial design and soil sampling, including the analytical methods for determination of carbonates and soil sorption capacity. We appreciate the reviewer's insightful comments regarding the statistical analysis in our manuscript. We made significant revisions to address the concerns raised. The reviewer correctly pointed out that the initial two-way ANOVA design may not be appropriate for an experiment with three factors, each with several levels. In response to this, we have incorporated a 3-way ANOVA, which accounts for the treatment, sampling depth, and sampling year. This analysis provides an evaluation of the experimental factors and their potential interactions. The results obtained from this extended analysis were found to be significant, particularly with respect to the main effect of the treatment, which was the aim of this study. We also improved results and discussion part, especially in the part of nutrient regime and harmful/hazardous substances. We hope that we have managed to meet all the expectations regarding the changes in the manuscript and that the clarity and interpretation of the results were improved. All the changes/additions in the revised paper are highlighted via the "track changes" tool.

Specific demands are:

-        Please, describe the spatial design of the experiment: Are you using a unique vineyard? If yes, there is pseudo replication in the design; how big is the vineyard or vineyards? Please clearly describe the spatial design of the experimental plots.

Response: Thank you for your suggestion. Improved, corrected, and demands accepted.

-        Please, describe correctly the “control” treatment, corresponding to the pre-experimental management of the vineyard.

Response: Thank you for your suggestion. Rewritten, corrected, and demand accepted.

-        Provide analyses of the organic materials used as manure.

Response: Analysis of poultry manure are added in manuscript (Experimental Seput in the Vineyard).

-        The statistical treatment is not clear: (a) the tables do not include control values; (2) sampling time -which year- is not specified; (c) are all sampling levels pooled? 

Response: Thank you for your suggestion. We appreciate the reviewer's attention to experimental details and would like to clarify that our experimental design intentionally did not include a control group. The primary focus of our study was to compare the various treatments directly against each other rather than against a control condition. Regarding the concern about specifying the sampling time, we would like to clarify that the experiment was conducted over the years 2019 to 2023. Regarding the concern about whether sampling levels were pooled, we would like to say that there were two distinct sampling levels in our experiment. In our revised manuscript, we explicitly stated the range of years during which the experiment took place.

-        Improve the introduction and create a discussion.

Response: Thank you for your suggestion. Improved, and demand accepted.

-        Review the text for some minor language mistakes.

Response: Thank you for your suggestion. Checked, and demand accepted.

Reviewer 2 Report

Comments and Suggestions for Authors

Dear Agronomy Editorial Office

I am sharing my observations about the manuscript “agronomy-2743651- Optimizing Soil Management for Sustainable Viticulture: Insights from a Rendzic Leptosol Vineyard in the Nitra Wine Region, Slovakia”. The manuscript studied the effects of soil management on the chemical attributes of soil at depths 0-30 cm and 30-60 cm. There were five treatments for soil management, but the authors didn’t write about experimental design. The authors wrote that they studied soil management and years, but they didn’t do interaction analysis or write about them. The results and discussion showed the chemical attributes as a function of soil management at different depths (0-30 cm and 30-60 cm). There were no analysis methods for “sorption capacity” and “carbonates”, but the results are in the manuscript.

Best regards!

21 November 2023

Author Response

Reviewer 2:

I am sharing my observations about the manuscript “agronomy-2743651- Optimizing Soil Management for Sustainable Viticulture: Insights from a Rendzic Leptosol Vineyard in the Nitra Wine Region, Slovakia”. The manuscript studied the effects of soil management on the chemical attributes of soil at depths 0-30 cm and 30-60 cm. There were five treatments for soil management, but the authors didn’t write about experimental design. The authors wrote that they studied soil management and years, but they didn’t do interaction analysis or write about them. The results and discussion showed the chemical attributes as a function of soil management at different depths (0-30 cm and 30-60 cm). There were no analysis methods for “sorption capacity” and “carbonates”, but the results are in the manuscript.

Response: We would like to thank the reviewer for going through our manuscript, feedback and constructive comments that helped us to improve our manuscript. We have tried to meet requests and suggestions. Comments in pdf document were incorporated into the revised manuscript as well. We appreciate the reviewer's careful consideration of our manuscript. We acknowledge the comment regarding the study of soil management and years without detailed analysis or discussion of their interactions. Upon revisiting our methodology and results, we recognize the importance of investigating interactions between soil management and years. In our revised manuscript, we conducted an interaction analysis by incorporating a 3-way ANOVA, which accounts for the treatment, sampling depth, and sampling date. This extended analysis provides an examination of the experimental factors and their potential interactions. We hope that we have managed to meet all the expectations regarding the changes in the manuscript and that the clarity and interpretation of the results were improved. All the changes/additions in the revised paper are highlighted via the "track changes" tool. We hope that we have met the expectations of the reviewer. 

Reviewer 3 Report

Comments and Suggestions for Authors

Several critical comments and remarks regarding the article by Simansky et al. “Optimizing soil management for sustainable viticulture: insights from a rendzic leptosol vineyard in the Nitra wine region, Slovakia”.

Understanding how different agrotechnical operations affect the bioavailability of hazardous elements is more pertinent and significant than studying changes in the movement of toxic elements within the soil profile. Therefore, the hazardous metal content must be determined not only in the soil but also in various plant organs (before and after the experiment). To determine the extent to which the mobility of harmful metals has changed after various soil treatment techniques have been applied, it is also necessary to compute the change in the transfer coefficient, which is calculated as the ratio of the metal concentration in different plant organs to that in soil. Please provide these data.

There are no arguments for selecting five metals for investigation (chromium, copper, nickel, lead, and zinc). The authors ascribe the elevated copper concentration in the soil to the application of copper- and zinc-based fungicides in the Results and Discussion section (lines 281-286). Provide information on fungicide usage in the research area. Brand names of the fungicides and their manufacturers. Their chemical composition. Provide information on fungicide doses and treatment frequency in the study region. Justify the selection of lead, nickel, and chromium as additional hazardous metals for the study.

“In the Slovak Republic, the permissible limit for hazardous elements in soils with different soil textures is regulated by Act 220/2004…” (Lines 271-275). All laws, rules, acts, and legislation must be disclosed in the public domain. Please provide a link to this document (Act 220/2004). Furthermore, this Act does not contain any information on the limitation of the content of any harmful elements in Slovakian soil, according to the national law database! (See: Zákon č. 220/2004 Z. z.Zákon o ochrane a využívaní poľnohospodárskej pôdy a o zmene zákona č. 245/2003 Z. z. o integrovanej prevencii a kontrole znečisťovania životného prostredia a o zmene a doplnení niektorých zákonov. Link: https://www.zakonypreludi.sk/zz/2004-220)

Because Agronomy is an international journal, most readers are unfamiliar with the Nitra wine region. Therefore, it would be a good idea to show a map of the location of the Nitra wine region inside Slovakia as well as the localisation of the sample plots within the research region.

Please enter the area size for each of the five different soil management practises. Also, provide the size of the sampling plots.

Add research limitations to the Discussion section.

The text contains the number 2 in two tables. Please revise the table numbering.

Author Response

Reviewer 3:

Several critical comments and remarks regarding the article by Simansky et al. “Optimizing soil management for sustainable viticulture: insights from a rendzic leptosol vineyard in the Nitra wine region, Slovakia”.

Response: We would like to thank the reviewer for going through our manuscript, feedback on the topic of the manuscript and constructive comments that helped us to improve our manuscript. All suggestions and comments were considered, and we have tried to modify the text and include the missing information. We hope that we have met expectations of the reviewer.

Understanding how different agrotechnical operations affect the bioavailability of hazardous elements is more pertinent and significant than studying changes in the movement of toxic elements within the soil profile. Therefore, the hazardous metal content must be determined not only in the soil but also in various plant organs (before and after the experiment). To determine the extent to which the mobility of harmful metals has changed after various soil treatment techniques have been applied, it is also necessary to compute the change in the transfer coefficient, which is calculated as the ratio of the metal concentration in different plant organs to that in soil. Please provide these data.

Response: Thank you for your suggestion and your opinion. Information about the amount of harmful/hazardous substances in the soil profile is very important, because it is possible to quantify their concentration in the soil and, based on this, it is possible to carry out remedial measures for their elimination/immobilization. Of course, it is also very important to deal with the transfer of harmful elements from the soil to plants and subsequently to the food chain. Unfortunately, we are currently not able to investigate this, because information about harmful elements in plant materials is not available. We did not solve this problem. Thanks for the topic for the future work. However, we tried to significantly improve the results and discussions in this section.

There are no arguments for selecting five metals for investigation (chromium, copper, nickel, lead, and zinc). The authors ascribe the elevated copper concentration in the soil to the application of copper- and zinc-based fungicides in the Results and Discussion section (lines 281-286). Provide information on fungicide usage in the research area. Brand names of the fungicides and their manufacturers. Their chemical composition. Provide information on fungicide doses and treatment frequency in the study region. Justify the selection of lead, nickel, and chromium as additional hazardous metals for the study.

Response: Thank you for your suggestion. We have added information about the used fungicides to the material and methodology section. We have also improved the description and discussion of hazardous element results.

“In the Slovak Republic, the permissible limit for hazardous elements in soils with different soil textures is regulated by Act 220/2004…” (Lines 271-275). All laws, rules, acts, and legislation must be disclosed in the public domain. Please provide a link to this document (Act 220/2004). Furthermore, this Act does not contain any information on the limitation of the content of any harmful elements in Slovakian soil, according to the national law database! (See: Zákon č. 220/2004 Z. z.Zákon o ochrane a využívaní poľnohospodárskej pôdy a o zmene zákona č. 245/2003 Z. z. o integrovanej prevencii a kontrole znečisťovania životného prostredia a o zmene a doplnení niektorých zákonov. Link: https://www.zakonypreludi.sk/zz/2004-220)

Response: Thank you for your suggestion. Added to the references.

Because Agronomy is an international journal, most readers are unfamiliar with the Nitra wine region. Therefore, it would be a good idea to show a map of the location of the Nitra wine region inside Slovakia as well as the localisation of the sample plots within the research region.

Response: Thank you for your suggestion. Figure 1 added.

Please enter the area size for each of the five different soil management practises. Also, provide the size of the sampling plots.

Response: Thank you for your suggestion. Improved, and demand accepted.

Add research limitations to the Discussion section.

Response: Thank you for your suggestion. Improved, and demand accepted.

The text contains the number 2 in two tables. Please revise the table numbering.

Response: Thank you for your suggestion. Checked, corrected and demand accepted.

Round 2

Reviewer 2 Report

Comments and Suggestions for Authors

Dear MDPI Agronomy Editorial Office

I am sharing my observations about the manuscript “agronomy-2743651- Optimizing Soil Management for Sustainable Viticulture: Insights from a Rendzic Leptosol Vineyard in the Nitra Wine Region, Slovakia”. The authors answered the doubts and changed the manuscript.

Author Response

Reviewer 2:

I am sharing my observations about the manuscript “agronomy-2743651- Optimizing Soil Management for Sustainable Viticulture: Insights from a Rendzic Leptosol Vineyard in the Nitra Wine Region, Slovakia”. The authors answered the doubts and changed the manuscript.

Response: We would like to thank the reviewer for going through our manuscript, and positive feedback. Manuscript was improved after second revision too. All the changes/additions in the revised paper are highlighted via the "track changes" tool. We hope that we have met the expectations of the reviewer.

Reviewer 3 Report

Comments and Suggestions for Authors

I appreciate that the authors have considered most of the comments and made significant changes to the manuscript.

However, the manuscript needs minor revisions.

The comment " Add research limitations to the Discussion section" was ignored. The information was not included in the Discussion section.

Author Response

Reviewer 3:

I appreciate that the authors have considered most of the comments and made significant changes to the manuscript. However, the manuscript needs minor revisions. The comment " Add research limitations to the Discussion section" was ignored. The information was not included in the Discussion section.

Response: We would like to thank the reviewer for going through our manuscript, feedback on the topic of the manuscript and constructive comments. All suggestions and comments were considered, and we have tried to modify the text and include the missing information. We stated the limitation of this study in the conclusions in revision 1. In revision 2, we stated the limitation of this study in the results and discussion section. Manuscript was improved after second revision too. All the changes/additions in the revised paper are highlighted via the "track changes" tool. We hope that we have met the expectations of the reviewer.
